# Effect of family support and work resources in the relationship of economic constraints and work volition: Evidence from China

**Lu Hai[1], Yang Wang[1]\*, Man Shu[2], Mengxiao Zhang[3], Yijiao Wang[1]**

**1** Department of Education, Minzu University of China, Beijing, China, **2** Department Of Sociology, Peking University, Beijing, China, **3** Department of Education, Northeast Normal University, Changchun, Jilin, China

\* 1164962363@qq.com

## Abstract

### Background

As the pace of economic development slows, college students are facing an increasingly challenging employment landscape. For instance, the expansion of higher education has led to a swell in the number of job seekers, which has in turn intensified competition. Given the limited job opportunities, it's understandable that many college students are developing a pessimistic employment mindset. Therefore, it's crucial to explore how objective factors influence their work aspirations. But few studies have explored the role of mediating factors between the two, such as family and resource factors. Thus, this study examines the effects of family support and work resources between the relationship between economic constraints and work volition.

### Methods

The study examined 1249 Chinese undergraduate students as participants ((714 men and 535 women; $M_{age}$ = 19.32, SD = 1.50), using the questionnaire with the Wenjuanxing online survey tool. The questionnaire were collected between August, 2022 and December, 2022. SPSS 21.0 and AMOS 24.0 were used to conducted a comprehensive analysis of the collected data, and investigate the relationships among latent variables and assess the goodness of fit of the observed indicators on their associated latent constructs. Additionally, we evaluated all the hypothesized direct and indirect effects.

### Results

The results showed the direct and indirect relationships among economic constraints, family support, work resources and work volition. Economic constraints can directly predict work volition. Moreover, economic constraints have a significant negative impact on work volition via two mediators: family support and work resources. On the one hand, economic constraints negatively affect work volition through family support and work resource separately. On the other hand, economic constraints negatively predict family support and work resource, thus negatively impact work volition.

**Data Availability Statement:** All relevant data are within the manuscript and its Supporting information files.

**Funding:** The authors received no specific funding for this work.

**Competing interests:** The authors have declared that no competing interests exist.

## Contribution

The current study has established the independent mediating and chain-mediated effects of family support and work resources on the relationship between economic constraints and work volition. This deeper understanding of internal mechanisms provides valuable insights that can inform strategies for enhancing individual's work volition, particularly from the perspectives of economic constraints, family support, and work volition.

## Introduction

In recent years, due to the economic downturn and the college expansion plan, the youth demographic, primarily comprised of university graduates, is encountering increasingly greater challenges in securing gainful employment [1]. Amidst such immense pressure, it is noteworthy that a strong work ethic can serve as a valuable asset in securing suitable employment opportunities in the future [2, 3].

Work volition is defined as an individual's sense of agency or freedom to make a career choice [4], highlighting the potential of subjective belief to overcome external constraints. For contemporary Chinese college students, their work volition exhibits two key characteristics. Firstly, overall, the level of work volition is generally low. Secondly, individuals from low-income families possess lower work volition than those from high socioeconomic status families [5]. Notably, economic constraints can negatively impact work volition, a finding consistently supported by some researches in different areas. It was found that individuals with greater economic constraints tend to have lower levels of career decision-making among university students and employees [6, 7]. Meanwhile, The PWT (Duffy et al., 2016) was developed to capture the work lives of all individuals. In PWT moedel, economic constraints and economic constraints are positioned as key predictors, in particular, economic constraints can objectively predict work volition. Additionally, economic constraints can also escalate conflicts and distress within families [8, 9], for example, young individuals from families who are economically disadvantaged tend to encounter both internal and external issues [10], undermining adolescents' access to family support. Given the strong emphasis on family values in Chinese culture, family support also plays a crucial role in individuals' growth and career development. Family support can bring emotional encouragement and work support to individuals, so that they feel more confident when they are making choice about work [11–15]. Furthermore, studies suggest that economic constraints are also associated with work resources [15], for instance, Duffy et al. (2016) suggested that economic constraints limit capital resources that can be applied in the workplace, making it more difficult for individuals in economic constraints to acquire the resources that privileged individuals take for granted [16]. And work resources are recognised as a fundamental requirement for individuals in career development [17].

Previous studies have examined the relationships among economic constraints, family support, work resources and work volition. It was demonstrated the direct effects of economic constraints on work volition and the direct impact of family support and work resources on work volition. However, there are still knowledge gaps. Obviously previous research has not fully considered the mediating role of objective factors in the relationships between economic constraints and work volition, such as family support and work resources. Therefore, it has not explored the internal mechanisms that economic constraints use to influence work volition through family support and work resources have not been fully explored.

Therefore, the goal of this study is to investigate the role played by family support and work resources in the relationship between economic constraints and work volition. We also aim to examine the complete pathway through which economic constraints indirectly affect work volition among Chinese college students facing employment issues.

## Economic constraints and work volition

Work volition refers to an individual's sense of agency or freedom in making career choices [4]. Previous studies have examined the relationship between work volition and various variables, including economic constraints [16, 18], academic satisfaction [19], social status, occupational engagement [20], social support [21], and others. Among these variables, economic constraints are considered to be the most significant.

Economic constraints refer to a lack of economic resources, which encompasses not only direct material goods but also access to economic resources [22, 23]. Cheung et al. (2020) and Tokar et al. (2018) examined different groups of subjects (workers and students) from different regions (United States and Hong Kong), and found that economic constraints were negatively related to work volition [24, 25]. Zhang et al. (2019) conducted a study among Chinese college students and found that students from high-income families had higher work volition than those from low-income families. Allen et al. (2020) examined the relationship between economic constraints and work volition from a longitudinal perspective, which further proved that economic constraints negatively affect work volition [6].

## Family support as the first mediating variable

Family support refers to the positive resources that individuals acquire from their family relationships [26], a dimension of social support that is a contextual factor known to influence career-related behavior. Many psychologists have emphasized the significance of family support in vocational development because it is directly linked to career search self-efficacy [27]. Family support can relieve personal stress so that they can have more energy to devote to their work [28]. Some studies have suggested that individuals supported by their families report greater work volition. For instance, Fouad et al. (2010) developed the Family Influence Scale (FIS) to examine the influence of family on career and work choices. They also found that family's informational support and financial support can impact career and work choices, which can be seen as work volition [15]; in other words, family support can predict a higher level of work volition. Other studies have suggested that family support is significantly related to career optimism [11], which involves making free work choices with a positive mindset despite obstacles, fitting the definition of work volition. According to some studies conducted on Chinese college students, those with adequate career-related parental support are more likely to feel a sense of control, curiosity, confidence about work, and are able to choose the work that best suits their needs, interests, and values. In other words, individuals with greater family support will have greater work volition [29–31].

Furthermore, research indicates that there is a negative correlation between financial constraints and familial support. This implies that families who are struggling financially are less likely to provide their offspring with monetary assistance or various forms of financial aid. For instance, economic stress can negatively impact parenting styles, where high-quality parenting offers children comprehensive support.

Based on these studies, we establish the first hypothesis:

**H1:** Family support mediates the relationship between economic constraints and work volition.

## Work resources as a second mediating variable

**Work resources** refer to the factors that stimulate personal growth, help achieve work goals, and mitigate the costs of job demands, which reflect physical, social, psychological, and organizational aspects [32]. They are intricately linked to experiences of job satisfaction, autonomy, purpose, engagement, meaningful work, and job performance [33]. Numerous empirical studies have found that when individuals lose resources at work, they are more likely to have negative physiological outcomes such as burnout and depression [34], which could adversely affects the work volition. According to Barbier et al. (2013), work and personal resources exert a causal effect on job engagement, which is a component of work volition [35]. Furthermore, individuals who lack essential work resources may feel burned out [36], making it more challenging to make informed job decisions, i.e., exhibiting a lower level of work volition. Cheung et al. (2020) examined the relationship between personal resources, constraints, and work volition among undergraduate students in the United States and Hong Kong. Their findings revealed a positive correlation between work resources and work volition, indicating that individuals with greater work resources are more likely to possess the capacity to make informed work decisions [24]. Choi et al. (2022) examined the relationship between social class and work volition, and their findings revealed a positive correlation between the two [37]. This supports the theory that individuals from lower social classes, who may face economic constraints, tend to exhibit lower levels of work volition. A study among Chinese college students revealed that those from high socioeconomic backgrounds, who have greater access to helpful work resources, tend to exhibit higher levels of work volition [38]. Additionally, research has shown that economic constraints can limit individuals' access to work resources. Bourdieu (2008) suggested that economic constraints limit individuals' access to these resources, as individuals or their families may need to pay for them, which can be difficult when facing economic constraints [39]. A study among labour market entrants found that individuals whose parents had higher education levels, were employed, or were from the service class (indicating greater economic income) had greater access to job-finding resources [40]. In other words, less economic constraint promotes greater access to job resources.

Based on these studies, we establish the second hypothesis:

**H2:** Work resources mediate the relationship between economic constraints and work volition.

## Family support and work resources

About the relationship between these two mediated variables, some studies indicated the relationship between these two mediating variables has been explored in several studies, which suggest that family support is associated with work resources. Several theories have demonstrated the relationship between family and work resources. Bourdieu's theory (2008) postulates that families with diverse capital can provide individuals with resources, including work-related resources such as a professional network, job opportunities, and a clear understanding of job rules [39]. Conservation of Resources theory raised by Hobfoll indicates that in order to prevent the loss of resources, individuals invest in resources and acquire the corresponding resources [41]. For college students, families make many investments in resources for their work. The family investment model suggests that families of higher socioeconomic status tend to leverage their financial and networking strengths for investment, which means more family support promotes more work resources [42]. Researchers have also conducted empirical studies to test this relationship. Faas et al. (2013) found that families with high occupational status

can provide their children with work resources such as professional training, communication skills, and managerial abilities [43]. In other words, individuals with various forms of family support are likely to have access to ample work resources. Wayne and Matthews et al. (2020) conducted a cross-sectional survey of employees and found that work resources are associated with lower work-to-family conflict [44], which suggests that individuals can receive greater family support for their work. Furthermore, a study focusing on college students revealed that higher levels of family cohesion were associated with career thoughts, an important aspect of work volition. This suggests that a harmonious family can provide greater support and promote a higher level of work volition [45].

Based on these studies and combine the findings that family support and work resource play important roles between economic constraints and work volition, we established the third hypothesis:

H3: Family support and work resources played a chain-mediating role in the relationship between economic constraints and work volition.

## Hypothesis model

The aim of this study was to examine the mechanisms underlying the relationship between economic constrains and work volition through the family support and work resources using a mediation model. In this research, three variables are positioned as key predictors of work volition: (a) economic constraints and (b) family support and (c) work resource as variables that both predict work volition and explain the mediating role of family support and work resource. Specifically, (1) family support mediates the relationship between economic constraints and work volition; (2) Work resources mediate the relationship between economic constraints and work volition; (3) Family support and work resources played a chain-mediating role in the relationship between economic constraints and work volition. Based on these three hypothesis pathways, we constructed the model (See Fig 1).

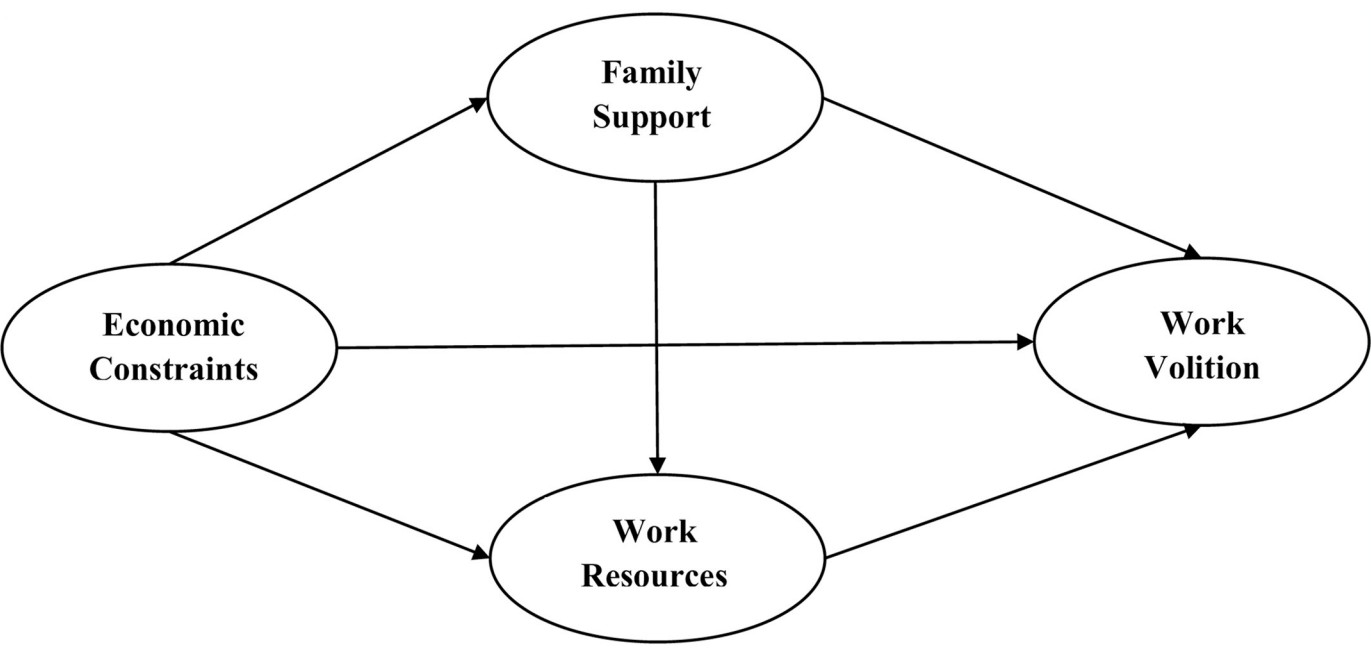

**Fig 1. Conceptual model.**

## Method

### Participants

To ensure the sample's representativeness, the questionnaires of this study are respectively distributed to 1–2 cities of eastern, central, and western regions of China, including Chongqing, Shandong, Anhui, Guangxi, and Jiangsu provinces. Participants in the present study were 1249 undergraduate students with a mean age of 19.32 ($SD$ = 1.50). There were 714 male (57.2%) and 535 female (42.8%) participants. Additionally, 598 (47.9%) were freshmen, 252 (20.2%) were sophomores, 206 (16.5%) were in their third year, and 193 (15.5%) were senior-year students. The participants were mainly from Chongqing, Shandong, Anhui, Guangxi, and Jiangsu provinces in China. Regarding monthly household income, participants reported 123 (9.8%), 301 (24.1%), 271 (21.7%), 193 (15.5%), 123 (9.8%), and 238 (19.1%), respectively, earned less than 2,000 RMB, 2,000–4,000 RMB, 4,000–6,000 RMB, 6,000–8000 RMB, 8,000–10,000 RMB, and more than 10,000 RMB.

### Procedure

After determining the target regions for questionnaire distribution, the research team provided detailed information about the questionnaire and shared the survey link with selected teachers in various provinces. These teachers were then tasked with distributing the questionnaires to students in their respective regions using WeChat, a widely utilized social media platform. This process was carried out between August 1st and December 31st, 2022. Following data collection, the research team organized and conducted a comprehensive analysis of the collected data. Specifically, SPSS 21.0 was utilized to calculate measures such as skewness, kurtosis, internal consistency reliability, and common method bias (CMB). Meanwhile, AMOS 24.0 was used to compute average variance extracted (AVE), composite reliability (CR), heterotrait-monotrait ratio of correlations (HTMT), and the structural equation model (SEM). This analysis aimed to investigate the relationships among latent variables and assess the goodness of fit of the observed indicators on their associated latent constructs. Additionally, it evaluated all the hypothesized direct and indirect effects.

### Ethics statement

Informed consent was obtained from all subjects involved in the study by written style. The participants provided consent by completing the questionnaire (https://www.wjx.cn/vm/Q0VNs6G.aspx#) in Wenjuanxing. Note at the beginning of the questionnaire: This questionnaire is not mandatory, you have the right to withdraw at any time, if you fill out and submit the questionnaire, it is deemed that you understand and agree to our use of your relevant data. Therefore, every participant is informed. Their participation was anonymized. The study was conducted in accordance with the Declaration of Helsinki and approved by the Ethics Committee of the School of Education, Minzu University of China (approval date: June 1, 2022).

### Instruments

**Economic constraints.** We measured the degree of economic constraints using a 5-item Economic constraints Scale [46]. The items are as follows: 1) "For as long as I can remember, I have had very limited economic or financial resources," 2) "I have faced financial dilemma most of my life,," 3) "For as long as I can remember, I have had difficulties making ends meet," 4) "I have considered myself poor or very close to poor most of my life," and 5) "For most of my life, I have not felt financially stable." All items were answered on a 7-point Likert scale

ranging from 1 (strongly disagree) to 7 (strongly agree). In the current study, the internal consistency reliability was α = 0.95.

**Family support.**  Family Support was assessed using the Multidimensional Scale of Perceived Social Support [47]. Sample items were "My family can help me in concrete ways" and "I can talk about my problems with my family." All items were answered on a 7-point Likert scale ranging from 1 (strongly disagree) to 7 (strongly agree). In this study, internal consistency reliability was α = 0.87.

**Work resources.**  Work resources were measured using the 3-item Access to Resources subscale of the Psychological Empowerment Scale [48]. Participants answered on a 5-point Likert scale ranging from 1 (strongly disagree) to 5 (strongly agree). Sample items are "I had access to the necessary resources to support my new ideas," "When additional resources are needed to do my job, I usually have them available," and "I have access to the resources I need to do my job well." In this study, the internal consistency coefficient of the scale was α = 0.90.

**Work volition.**  Work volition was measured using the 7-item Work Volition Scale [49]. Sample items include, "I can do the kind of work I want, despite external barriers," "I will learn how to find my own way in the world of work," and "I feel total control over my future job choices." Items were answered on a 7-point Likert scale ranging from 1 (strongly disagree) to 7 (strongly agree). In this study, after removing two items based on the results, there are five items remaining, the internal consistency coefficient of the scale was α = 0.84.

## Results

### Preliminary analyses

Before analyzing the data, we conducted preliminary analyses using IBM SPSS 21.0. The skewness and kurtosis of all variables were between -0.66 and 0.27 and between −0.78 and 0.33, respectively. On account of the absolute values of skewness ≤ 3.0 and kurtosis ≤ 10.0 [50], the shape of the data distribution in the current study may not be severely non-normal. All items in the questionnaire were subjected to Harman's single-factor test, and the results showed that the first factor explained 36.11% of the total variance, which is below the critical value of 40%. This indicates that there is no significant presence of common method bias.

Moreover, the results showed that the CR of the four scales were 0.95, 0.88, 0.90 and 0.84 respectively, denoting high composite reliability based on the principle of Hair et al. (1998) [51]. The results showed that the AVE of the four scales were 0.79, 0.65, 0.75 and 0.52, and the square root of the number AVE on the diagonal of the scale mostly higher than the Pearson correlation among scales, indicating that these questionnaire have strong convergent validity and discriminant validity based on the principle of Fornell and Larcker(1981) [45]. And, the results showed that the HTMT between the four scales were less than 0.85, further denoting high composite reliability based on the principle of Henseler, Ringle, & Sarstedt(2015) [52]. The descriptive statistics also are presented (see Table 1).

### Measurement model

We constructed measurement models to examine correlations among latent variables and the goodness of fit of the observed indicators on their associated latent constructs with Amos Graphics 24.0; we found $\chi^2$/df = 4.78, RMSEA = 0.06, TLI = 0.97, CFI = 0.97, and SRMR = 0.04, indicating a good fit according to some thresholds. We, then, tested these structural models. The factor correlations and descriptive statistics are presented in Table 1.

**Table 1. Descriptive statistics and correlations of variables.**

| | CR | AVE | 1 | 2 | 3 | 4 |
|---|---|---|---|---|---|---|
| 1. Economic constraints | 0.95 | 0.79 | 0.89 | 0.24 | 0.22 | 0.20 |
| 2. Family Support | 0.88 | 0.65 | -0.22** | 0.81 | 0.51 | 0.45 |
| 3. Work Resources | 0.90 | 0.75 | -0.21** | 0.45** | 0.87 | 0.63 |
| 4. Work Volition | 0.84 | 0.52 | -0.18** | 0.38** | 0.55** | 0.72 |
| M | | | 3.78 | 5.14 | 3.50 | 4.84 |
| SD | | | 1.50 | 1.15 | 0.76 | 1.01 |

Note:

$^*p < 0.05$,

$^{**}p < 0.01$,

$^{***}p < 0.001$.

The data on the diagonal of different questionnaires represent the square root of AVE. The data below the diagonal represent the correlation coefficients between different questionnaires. The data above the diagonal represent the HTMT values of the questionnaires.

## Structural model

We tested structural models that contained all the hypothesized direct and indirect effects and found that most paths were significant. The results of $\chi^2$/df = 4.78, RMSEA = 0.06, TLI = 0.97, CFI = 0.97, and SRMR = 0.04 indicate a good fit according to some thresholds. In terms of direct paths, economic constraints failed to predict work volition directly (β = -0.05, SE = 0.03, p = 0.13, 95% CI [-0.11, 0.01]). Economic constraints had significant direct effects on family support (β = -0.21, SE = 0.03, p <0.001, 95% CI [-0.27, -0.14]) and work resources (β = -0.11, SE = 0.03, p <0.01, 95% CI [-0.18, -0.05]); family support had significant direct effects on work resources (β = 0.47, SE = 0.03, p <0.001, 95% CI [0.40, 0.53]) and work volition (β = 0.15, SE = 0.04, p <0.001, 95% CI [0.07, 0.23]); work resources had significant direct effects on work volition (β = 0.53, SE = 0.04, p <0.001, 95% CI [0.46, 0.60]).

As for the indirect paths, economic constraints had a significant indirect effect on work volition via family support and work resources (β = -0.02, SE = 0.01, p <0.001, 95% CI [-0.03, -0.01] and β = -0.04, SE = 0.01, p <0.01, 95% CI [-0.06, -0.02], respectively). Additionally, we examined a chain mediation that economic constraints to work volition via family support and work resources (β = -0.03, SE = 0.01, p <0.001, 95% CI [-0.05, -0.02]). All paths are listed in Tables 2 and 3.

**Table 2. Paths coefficient from structural model.**

| Paths | β | SE | 95% confidence interval bootstrap bias corrected | |
|---|---|---|---|---|
| | | | Lower bound | Upper bound |
| EC→FS | -0.21*** | 0.03 | -0.27 | -0.14 |
| EC→WR | -0.11** | 0.03 | -0.18 | -0.05 |
| EC→WV | -0.05 | 0.03 | -0.11 | 0.01 |
| FS→WR | 0.47*** | 0.03 | 0.40 | 0.53 |
| FS→WV | 0.15*** | 0.04 | 0.07 | 0.23 |
| WR→WV | 0.53*** | 0.04 | 0.46 | 0.60 |

Note: EC = economic constraints; FS = family support; WR = work resources; WV = work volition.

$^*p < 0.05$,

$^{**}p < 0.01$,

$^{***}p < 0.001$.

**Table 3. Direct and indirect effects from the structural model.**

| paths | effect type | β | SE | 95% confidence interval bootstrap bias corrected | | Ratio |
| --- | --- | --- | --- | --- | --- | --- |
| | | | | Lower bound | Upper bound | |
| EC→FS/WR/FS&WR→WV | Direct effect | -0.03 | 0.02 | -0.07 | 0.01 | 0.60/0.44/0.50 |
| EC→FS→WV | Total effcet | -0.05* | 0.02 | -0.09 | -0.01 | - |
| | Indirect effcet | -0.02*** | 0.01 | -0.03 | -0.01 | 0.40 |
| EC→WR→WV | Total effcet | -0.06** | 0.01 | -0.11 | -0.02 | - |
| | Indirect effcet | -0.04** | 0.01 | -0.06 | -0.02 | 0.56 |
| EC→FS→WR→WV | Total effcet | -0.06* | 0.02 | -0.10 | -0.02 | - |
| | Indirect effcet | -0.03*** | 0.01 | -0.05 | -0.02 | 0.50 |

Note: EC = economic constraints; FS = family support; WR = work resources; WV = work volition.

*$p < 0.05$,

**$p < 0.01$,

***$p < 0.001$.

## Discussion

This study aims to test the effect of family support and work resources on the relationship between economic constraints and work volition using the data collected from college students in China. Overall, the results confirmed the hypotheses that economic constraints have a significant negative impact on work volition *via* two psychological mediators: family support and work resources.

Regarding the first hypothesis, economic constraints negatively affected work volition through family support and the ratio of indirect effcet is 0.49, which means family support plays an important meditating role. Specifically, those who suffered economic hardships received less family support, resulting in a lower level of ability to make career-related decisions, which is consistent with the findings of previous studies. For example, it is showed that students with economic constraints are more likely to face a variety of problems, which is not conducive for work volition [10]. Kim et al. (2018) also found that those who felt higher family support had more work volition than those who did not [20]. The conclusions can be explained by the following reasons. First, for students with financial difficulties, their parents spend most of their time working hard to earn money to meet their needs for necessary expenses; thus, these families do not have extra time or energy to provide abundant various support, both physically and emotionally. Second, people with more family support have more stress resistance; therefore, they have enough confidence and courage to make the best career choices without being afraid to take risks of failure.

Regarding the second hypothesis, economic constraints significantly predict work volition through work resources, and the ratio of indirect effcet is 0.63, which means work resources have a significant meditating effect. To elaborate, students in bad economic conditions are less likely to have access to ample work resources, and they will feel less freedom in their career choice. This result is consistent with the theory proposed by Bourdieu (2008) [39], which indicates that among the three guises of capital existing in the world (i.e., economic, cultural, and social), both cultural and social capitals can be derived from economic capital. This implies that economic constraints limit the gain of other types of capital, particularly work resources, in our case. Moreover, it is also in line with the Conversation of Resources theory [53], the core of stress is the fear of loss of resources; that is, people with more work resources are more likely to resist various work stresses. Therefore, they can perceive the freedom to choose a

career regardless of any potential stress or challenge. The findings can be attributed some explanations. First, the primary reason may be that most access to career resources, such as taking job training classes or consulting with job agencies, requires payment; however, people with economic difficulties do not have enough money to bear these costs. Second, students with abundant work resources usually acquire the professional knowledge and skills required for their work. These students always have a clear career plan that can partially increase their level of work volition.

Regarding the third hypothesis, family support and work resources act as a chain intermediary between economic constraints and work volition, and the ratio of indirect effcet is 0.60, which means both family support and work resources have important meditating effect. Specifically, economic constraints negatively predict family support. Family support has a significant positive effect on work resources, and work resources also have a significant positive effect on work volition. The relationships between economic constraints and family support and work resources and volition have been discussed above; here, we have discussed the relationship between family support and work resources. This finding indicates that people with adequate family support acquire better work resources, which is consistent with the findings of previous studies. It was found that interpersonal network is very important to people's life and career because it will bring valuable information which can not be acquired through one's own efforts [54]. For college students, most of their related-work interpersonal network is supported by their family. Thus, with the more family support, they are more likely to possess further information, such as vocational information, employment opportunities, and necessary encouragement, which can bring them better work resources. In sum, combined with the theory raised by Bourdieu (2008) that families in an economic dilemma are difficult to receive sufficient support [39], it is confirmed that those who have higher economic constrains perceive less family support, and lacking family support further limits the gain of work resource and leads to lower work volition.

## Contribution

The results of this study offer valuable insights into the roles played by family support and work resources in the processes linking economic strain to work volition. Specifically, having adequate family support and work resources can enhance the work volition of individuals facing economic difficulties. Therefore, our findings suggest potential avenues for targeted interventions among college students facing economic constraints, with the aim of boosting their work volition and promoting more effective career development.

First, based on the finding that limited economic conditions directly undermine the level of family support and work resources and work volition of university students. Governments and colleges are encouraged to establish specialized scholarships for those who are constrained by economic burden; thus, they will spend more time considering future career plans instead of worrying about their livelihood. In addition, vocational training and educational institutions are expected to offer free or discounted courses for students to receive vocational certificates, which could provide them with better career options. Moreover, recruitment companies should set up more paid internship opportunities for undergraduates in order to enrich their work resources. Therefore, they will be more experienced when making work choices.

Second, our findings suggest that family support is a salient factor that affects work resources and work volition. Therefore, more active strategies are required to promote family support. Parents are ought to provide their children with substantial support; for example, they can give their children funds to take part in lessons in order to get internship opportunities to improve their work-related beliefs and actual abilities. Families can also offer emotional

support. Specifically, parents should try to understand and respect their children's personal career choices, democratically raise appropriate suggestions, and encourage their children to explore their future without burden.

Finally, the mediating effects of work resources can inform appropriate interventions. Universities or vocational organizations are supported to disseminate considerable career-related information, such as development trends and hotspots of various professions. More experiential training and internships should also be offered, allowing students to gain experience and master skills matching their future work, which will increase their confidence in making work choices.

This study adds to the literature of the relationship between economic constrains and work volition by validating family support and work resource as novel mediating roles. Under the realistic background of college students' difficult employment, future research is required to further explore the internal mechanism of economic influence on work volition. We believe that the data from the present study could provide future research useful insights into understanding specific impacts at the family support and resource level.

## Limitations and future research directions

This study exhibits several limitations that probably provide directions for future research. First, we only investigated family support as a whole and did not examine it from specific dimensions (e.g., emotional and information support). Future research should focus on examining the role of specific aspects of family support in the occupational field to provide a more nuanced occupational picture. Second, the cross-sectional design was adopted for 1249 undergraduate students, which means casual relationships between the variables are difficult to confirm in a strict sense, and longitudinal designs can be used in future relevant studies. Finally, we focused on the effects of external objectives (e.g., economic constraints, family support, and work resources) on work volition and neglected internal psychological factors. Future research should focus on the effects of internal factors on work volition to explore these factors more comprehensively.

## Conclusion

The present study, which focused on Chinese college students, delved into the relationship between economic constraints and work volition, taking into account the mediating effects of family support and work resources. The findings confirmed the mediating roles and chain mediating roles of these factors in the relationship. Specifically, three indirect pathways were identified: family support mediates the relationship between economic constraints and work volition; work resources mediate the relationship between economic constraints and work volition; family support and work resources played a chain-mediating role in the relationship between economic constraints and work volition. While acknowledging several limitations, the study provides valuable directions for future research on the importance of family support and work resources in career development. It also underscores the need for schools, families, and society to recognize and act on these factors to promote positive career outcomes for students.

## Supporting information

**S1 Dataset.**
(XLS)

## Author Contributions

**Data curation:** Lu Hai.

**Methodology:** Yang Wang, Man Shu.

**Resources:** Lu Hai.

**Supervision:** Lu Hai.

**Writing – original draft:** Yang Wang, Man Shu, Mengxiao Zhang, Yijiao Wang.

**Writing – review & editing:** Yang Wang, Mengxiao Zhang.

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
