## [Decision Letter · Decision Letter 0]

13 Nov 2023

PONE-D-23-23939Effect of Family Support and Work Resources in the Relationship of Economic Constraint and Work Volition : Evidence from ChinaPLOS ONE

Dear Dr. Wang,

Thank you for submitting your manuscript to PLOS ONE. After careful consideration, we feel that it has merit but does not fully meet PLOS ONE’s publication criteria as it currently stands. Therefore, we invite you to submit a revised version of the manuscript that addresses the points raised during the review process.

We look forward to receiving your revised manuscript.

Kind regards,

Henri Tilga, PhD

Academic Editor

PLOS ONE

Journal Requirements:

**Additional Editor Comments:**

The Reviewers have provided several useful comments to increase the quality of this manuscript. Please carefully follow all the comments made by the Reviewers and revise the manuscript accordingly.

Reviewers' comments:

Reviewer's Responses to Questions

**Comments to the Author**

1. Is the manuscript technically sound, and do the data support the conclusions?

Reviewer #1: Yes

Reviewer #2: Yes

2. Has the statistical analysis been performed appropriately and rigorously? 

Reviewer #1: Yes

Reviewer #2: No

3. Have the authors made all data underlying the findings in their manuscript fully available?

Reviewer #1: Yes

Reviewer #2: No

4. Is the manuscript presented in an intelligible fashion and written in standard English?

Reviewer #1: No

Reviewer #2: Yes

5. Review Comments to the Author

Reviewer #1: This paper is related to an interesting topic, namely Effect of Family Support and Work Resources in the Relationship of Economic Constraint and Work Volition : Evidence from China for PLOS ONE

However, the manuscript needs to be further developed in order to meet the expected academic requirements.

The introductory section of the document lacks the requisite depth and detail. It is essential to provide a more comprehensive introduction, one that not only sets the stage for the study but also explicitly identifies the existing knowledge gaps that this research aims to address. Without a clear articulation of these gaps in the introduction, the readers may find it challenging to understand the context and significance of the research.

The segment dedicated to the literature review and hypotheses development exhibits some weaknesses. To bolster the quality of this section, it is imperative to delve into previous theories and research within the field. This entails a thorough discussion of the existing body of knowledge, highlighting key theories, findings, and their implications. Furthermore, it is crucial to elucidate the novel contribution of this research and provide a comprehensive explanation of how the model underpinning the study was constructed. The absence of this information leaves the model without a solid foundation, undermining the credibility and robustness of the research.

The summary or conclusion of the study appears to be overly succinct and leaves some key elements unaddressed. A comprehensive conclusion should not only summarize the key findings but also provide answers to the research questions posed at the outset of the study. By failing to address these questions in the conclusion, the research may be perceived as incomplete and lacking in closure. A more comprehensive summary and conclusion would help readers to better grasp the implications and significance of the study's findings.

Reviewer #2: Dear Author

Thank you for submitting your paper "Effect of Family Support and Work Resources in the Relationship of Economic Constraint and Work Volition : Evidence from China" to this journal "Plos One". I read your paper and gave my concern as follows:

1. The abstract is not properly written. The background, and contributions of the paper must be included.

2. Introduction section lacks the rationale of writing this paper. The paper must craft the research problem and essence of the problem to be addressed. Moreover, the contributions of the authors must be justified through a solid research gap that has not yet addressed.

3. Literature review of the paper or the developing of the hypothesis was based on very old literature. The author must develop or justify those relationships based on recent papers/citations.

4. In methodology, the authors did not ground this paper. Particularly, the research design, sampling design, sampling frame, data collection procedure, etc were not sufficiently explained.

5. Measurement model was not supported with the required estimates. Particularly, I don't observed AVE, CR, and HTMT issues in this paper.

6. Where is CMB reporting.

7. A separate section on the strengths or contributions must be added after discussion.

8. Notably, the authors ignored recent papers to cite in their paper. I will ask them to search and cite papers from recent years (particularly, 2021-2023).

Wish you all the best.

6. PLOS authors have the option to publish the peer review history of their article (what does this mean?). If published, this will include your full peer review and any attached files.

Reviewer #1: No

Reviewer #2: No

---

## [Author Response · Author response to Decision Letter 0]

7 Jan 2024

Dear Dr. Henri Tilga

Thank you very much for the opportunity to revise our manuscript entitled “Effect of Family Support and Work Resources in the Relationship of Economic Constraint and Work Volition : Evidence from China”. We would like to thank you and the reviewers for their helpful comments. The manuscript has been carefully revised according to the comments raised by the reviewers. Please find a point-to-point response to reviewers’ comments below. Changes to the manuscript are highlighted in blue. 

We believe that the revisions prompted by the reviewers’ comments have strengthened our manuscript. We hope it is now suitable for publication and look forward to hearing from you in course. 

Yours faithfully

Academic Editor's Comments to Author:

1.Please ensure that your manuscript meets PLOS ONE's style requirements, including those for file naming. 

AUTHORS’ RESPONSE: Thank you very much for your review of our manuscript and very helpful comments. We have made revisions according to your comments. We have meet PLOS ONE's style requirements. As for the segment of “Abstract”, we have added the “Background” and “Contribution”. (pp. 2, marked in blue):

Abstract

Background 

As the pace of economic development slows, college students are facing an increasingly challenging employment landscape. For instance, the expansion of higher education has led to a swell in the number of job seekers, which has in turn intensified competition. Given the limited job opportunities, it's understandable that many college students are developing a pessimistic employment mindset. Therefore, it's crucial to explore how objective factors influence their work aspirations. But few studies have explored the role of mediating factors between the two, such as family and resource factors.Thus, this study examines the effects of family support and work resources between the relationship between economic constraints and work volition. 

Methods

The study examined 1249 Chinese undergraduate students as participants ((714 men and 535 women; Mage = 19.32, SD = 1.50), using the questionnaire with the Wenjuanxing online survey tool. The questionnaire were collected between August, 2022 and December, 2022. 

Results

The results showed that economic constraints have a significant negative impact on work volition via two mediators: family support and work resources. On the one hand, economic constraints negatively affect work volition through family support and work resource separately. On the other hand, economic constraints negatively predict family support and work resource, thus negatively impact work volition. 

Contribution

The current study has established the independent mediating and chain-mediated effects of family support and work resources on the relationship between economic constraints and work volition. This deeper understanding of internal mechanisms provides valuable insights that can inform strategies for enhancing individual's work volition, particularly from the perspectives of economic constraints, family support, and work volition.

Moreover, we have also corrected the format of the references. In the original article, we used superscripts in roman numerals. After we refer to the format of other articles published in the PlOSONE, we use the notation of Arabic numerals. For example, in the original text is was “family support refers to the positive resources that people acquired from their family relationships (Betz, 1989)ⅲ” But in the revised article, it is corrected “family support refers to the positive resources that people acquired from their family relationships[3].”

In addition, we revised file naming which is meet journal requirements, including “Response to Reviewers”, “ Revised Manuscript with Track Change Manuscript” and “Manuscript”.

2.Please provide additional details regarding participant consent. In the ethics statement in the Methods and online submission information, please ensure that you have specified what type you obtained (for instance, written or verbal, and if verbal, how it was documented and witnessed). If your study included minors, state whether you obtained consent from parents or guardians. If the need for consent was waived by the ethics committee, please include this information.

AUTHORS’ RESPONSE: Thank you for raising these important issues. To ensure the sample’s representativeness, the questionnaires of this study are respectively distributed to 1-2 cities of eastern, central, and western regions of China, including Chongqing, Shandong, Anhui, Guangxi, and Jiangsu provinces. After determining the target regions for questionnaire distribution, the research team provided detailed information about the questionnaire and shared the survey link with selected teachers in various provinces. These teachers were then tasked with distributing the questionnaires to students in their respective regions using WeChat, a widely utilized social media platform. This process was carried out between August 1st and December 31st, 2022. Informed consent was obtained from all subjects involved in the study by written style. The participants provided consent by completing the questionnaire, and their participation was anonymized. The study was conducted in accordance with the Declaration of Helsinki and approved by the Ethics Committee of the School of Education, Minzu University of China (approval date: June 1, 2022). (pp. 7-8, marked in blue):

Method

Participants

To ensure the sample’s representativeness, the questionnaires of this study are respectively distributed to 1-2 cities of eastern, central, and western regions of China, including Chongqing, Shandong, Anhui, Guangxi, and Jiangsu provinces. Participants in the present study were 1249 undergraduate students with a mean age of 19.32 (SD = 1.50). There were 714 male (57.2%) and 535 female (42.8%) participants. Additionally, 598 (47.9%) were freshmen, 252 (20.2%) were sophomores, 206 (16.5%) were in their third year, and 193 (15.5%) were senior-year students. The participants were mainly from Chongqing, Shandong, Anhui, Guangxi, and Jiangsu provinces in China. Regarding monthly household income, participants reported 123 (9.8%), 301 (24.1%), 271 (21.7%), 193 (15.5%), 123 (9.8%), and 238 (19.1%), respectively, earned less than 2,000 RMB, 2,000–4,000 RMB, 4,000–6,000 RMB, 6,000–8000 RMB, 8,000–10,000 RMB, and more than 10,000 RMB.

Procedure

After determining the target regions for questionnaire distribution, the research team provided detailed information about the questionnaire and shared the survey link with selected teachers in various provinces. These teachers were then tasked with distributing the questionnaires to students in their respective regions using WeChat, a widely utilized social media platform. This process was carried out between August 1st and December 31st, 2022. Following data collection, the research team organized and conducted a comprehensive analysis of the collected data. Specifically, SPSS 21.0 was utilized to calculate measures such as skewness, kurtosis, internal consistency reliability, and common method bias (CMB). Meanwhile, AMOS 24.0 was used to compute average variance extracted (AVE), composite reliability (CR), heterotrait-monotrait ratio of correlations (HTMT), and the structural equation model (SEM). This analysis aimed to investigate the relationships among latent variables and assess the goodness of fit of the observed indicators on their associated latent constructs. Additionally, it evaluated all the hypothesized direct and indirect effects.

Ethics statement 

Informed consent was obtained from all subjects involved in the study by written style. The participants provided consent by completing the questionnaire, and their participation was anonymized. The study was conducted in accordance with the Declaration of Helsinki and approved by the Ethics Committee of the School of Education, Minzu University of China (approval date: June 1, 2022).

3.In your Data Availability statement, you have not specified where the minimal data set underlying the results described in your manuscript can be found. PLOS defines a study's minimal data set as the underlying data used to reach the conclusions drawn in the manuscript and any additional data required to replicate the reported study findings in their entirety. All PLOS journals require that the minimal data set be made fully available. For more information about our data policy, please see http://journals.plos.org/plosone/s/data-availability.

Upon re-submitting your revised manuscript, please upload your study’s minimal underlying data set as either Supporting Information files or to a stable, public repository and include the relevant URLs, DOIs, or accession numbers within your revised cover letter.

AUTHORS’ RESPONSE: Thank you for raising these important issues. We have uploaded our study’s minimal underlying data set as either Supporting Information files, which is named “data set”. The data that support this study are available from the corresponding author upon reasonable request.

4.PLOS requires an ORCID iD for the corresponding author in Editorial Manager on papers submitted after December 6th, 2016. Please ensure that you have an ORCID iD and that it is validated in Editorial Manager. To do this, go to ‘Update my Information’ (in the upper left-hand corner of the main menu), and click on the Fetch/Validate link next to the ORCID field. 

AUTHORS’ RESPONSE: Thank you for raising these important issues. We have provided ORCID ID for the corresponding author in Editorial Manager on papers submitted.

Reviewers’ Comments to Author:

Reviewer: 1

This paper is related to an interesting topic, namely Effect of Family Support and Work Resources in the Relationship of Economic Constraint and Work Volition : Evidence from China for PLOS ONE. However, the manuscript needs to be further developed in order to meet the expected academic requirements.

AUTHORS’ RESPONSE: Thank you very much for your review of our manuscript and very helpful comments. We have made revisions according to your comments.

1.The introductory section of the document lacks the requisite depth and detail. It is essential to provide a more comprehensive introduction, one that not only sets the stage for the study but also explicitly identifies the existing knowledge gaps that this research aims to address. Without a clear articulation of these gaps in the introduction, the readers may find it challenging to understand the context and significance of the research.

AUTHORS’ RESPONSE: Thank you for raising this issue. We very much agree with your opinion and do find that the original article did not provide a comprehensive introduction regarding a clear articulation of existing knowledge gaps. We have added the following content according to your suggestion (pp. 3, marked in blue):

Previous studies have examined the direct effects of economic constraints on work volition and the direct impact of family support and work resources on work volition. However, there are still knowledge gaps. For instance, previous research has not fully considered the mediating role of objective factors within the family and work in the relationship between family support and work volition, such as family support and work resources. Additionally, the internal mechanisms that economic constraints use to influence work volition through family support and work resources have not been fully explored.

Therefore, the goal of this study is to investigate the role played by family support and work resources in the relationship between economic constraints and work volition. We also aim to examine the complete pathway through which economic constraints indirectly affect work volition among Chinese college students facing employment issues.

2.The segment dedicated to the literature review and hypotheses development exhibits some weaknesses. To bolster the quality of this section, it is imperative to delve into previous theories and research within the field. This entails a thorough discussion of the existing body of knowledge, highlighting key theories, findings, and their implications. Furthermore, it is crucial to elucidate the novel contribution of this research and provide a comprehensive explanation of how the model underpinning the study was constructed. The absence of this information leaves the model without a solid foundation, undermining the credibility and robustness of the research.

AUTHORS’ RESPONSE: Thank you for raising these important issues. 

(1)As for the literature review and hypotheses development, in the original article, there are some very old literature and the the number of literature is also too low. Thus, for the segment of “Introduction” “Economic Constraints and Work Volition” “Economic Constraints and Work Volition” “Work resources as a second mediating variable” “Work resources as a second mediating variable”,we have increased the number of literature, and deleted some old ones and added many new ones in recent years, particularly 2021-2023. Moreover, we have delved into previous theories and researches, highlighting key theories, findings, and their implications (pp. 3-7, marked in blue):

Economic Constraints and Work Volition

Work volition refers to an individual's sense of agency or freedom in making career choices [4]. Previous studies have examined the relationship between work volition and various variables, including economic constraints [17][18], academic satisfaction [6][19], social status [26][21], occupational engagement [20], social support [21], and others. Among these variables, economic constraints are considered to be the most significant.

Economic constraints refer to a lack of economic resources, which encompasses not only direct material goods but also access to economic resources [22][23]. Some studies conducted in different areas have found that individuals with greater economic constraints tend to have lower levels of career decision-making among university students and employees [7]. Tokar et al. (2018) and Cheung et al. (2020) examined different groups of subjects (workers and students) from different regions (United States and Hong Kong), and found that economic constraints were negatively related to work volition [24][25]. Zhang et al. (2019) conducted a study among Chinese college students and found that students from high-income families had higher work volition than those from low-income families. Allen et al. (2020) examined the relationship between economic constraints and work volition from a longitudinal perspective, which further proved that economic constraints negatively affect work volition [6].

Family support as the first mediating variable

Family support refers to the positive resources that individuals acquire from their family relationships [27], a dimension of social support that is a contextual factor known to influence career-related behavior. Many psychologists have emphasized the significance of family support in vocational development because it is directly linked to career search self-efficacy [28].Some studies have suggested that individuals supported by their families report greater work volition. For instance, Founad et al. (2015) developed the Family Influence Scale (FIS) to examine the influence of family on career and work choices. They also found that family's informational support and financial support can impact career and work choices, which can be seen as work volition [15]; in other words, family support can predict a higher level of work volition. Other studies have suggested that family support is significantly related to career optimism[11], which involves making free work choices with a positive mindset despite obstacles, fitting the definition of work volition. According to some studies conducted on Chinese college students, those with adequate career-related parental support are more likely to feel a sense of control, curiosity, confidence about work, and are able to choose the work that best suits their needs, interests, and values. In other words, individuals with greater family support will have greater work volition [29][30][31]. 

Furthermore, research indicates that there is a negative correlation between financial constraints and familial support. This im

---

## [Decision Letter · Decision Letter 1]

23 Jan 2024

PONE-D-23-23939R1Effect of Family Support and Work Resources in the Relationship of Economic Constraint and Work Volition : Evidence from ChinaPLOS ONE

Dear Dr. Wang,

Thank you for submitting your manuscript to PLOS ONE. After careful consideration, we feel that it has merit but does not fully meet PLOS ONE’s publication criteria as it currently stands. Therefore, we invite you to submit a revised version of the manuscript that addresses the points raised during the review process.

We look forward to receiving your revised manuscript.

Kind regards,

Henri Tilga, PhD

Academic Editor

PLOS ONE

**Additional Editor Comments:**

The Reviewers have provided several useful and important comments to increase the quality of this manuscript. Please carefully follow all the comments made by the Reviewers and revise the manuscript accordingly.

Reviewers' comments:

Reviewer's Responses to Questions

**Comments to the Author**

1. If the authors have adequately addressed your comments raised in a previous round of review and you feel that this manuscript is now acceptable for publication, you may indicate that here to bypass the “Comments to the Author” section, enter your conflict of interest statement in the “Confidential to Editor” section, and submit your "Accept" recommendation.

Reviewer #1: (No Response)

Reviewer #2: All comments have been addressed

2. Is the manuscript technically sound, and do the data support the conclusions?

Reviewer #1: (No Response)

Reviewer #2: Partly

3. Has the statistical analysis been performed appropriately and rigorously? 

Reviewer #1: (No Response)

Reviewer #2: No

4. Have the authors made all data underlying the findings in their manuscript fully available?

Reviewer #1: (No Response)

Reviewer #2: Yes

5. Is the manuscript presented in an intelligible fashion and written in standard English?

Reviewer #1: (No Response)

Reviewer #2: Yes

6. Review Comments to the Author

Reviewer #1: While the article holds certain research value, it falls short in delivering compelling theoretical discussions. A well-developed theoretical foundation is crucial for any academic work. Moreover, the paper tends to make numerous assumptions and addresses a wide array of research questions, lacking the necessary focus. It is suggested that the author refines the research questions for a more coherent presentation. In all, the current manuscript is not well-grounded in the related literature, nor does it develop a theoretical framework that is novel enough for the journal readers.

Reviewer #2: Dear Authors

Thank you for submitting your revised draft. I read your draft and gave my comments down here.

1. I appreciate that you have gave significant efforts to improve this draft.

2. I found some errors in your citations.

3. I found that in figure 1 there are two diagrams which are seemed to be similar.

4. AVE of work volition is less than .50. You might go for item deletion to make it above .50 as it seems a serious threat to convergent validity.

Thank you.

7. PLOS authors have the option to publish the peer review history of their article (what does this mean?). If published, this will include your full peer review and any attached files.

Reviewer #1: No

Reviewer #2: No

---

## [Author Response · Author response to Decision Letter 1]

9 May 2024

PONE-D-23-23939

Effect of Family Support and Work Resources in the Relationship of Economic Constraint and Work Volition : Evidence from China

PLOS ONE

Dear Dr. Henri Tilga

Thank you very much for the opportunity to revise our manuscript entitled “Effect of Family Support and Work Resources in the Relationship of Economic Constraint and Work Volition : Evidence from China”. We would like to thank you and the reviewers for their helpful comments. The manuscript has been carefully revised according to the comments raised by the reviewers. Please find a point-to-point response to reviewers’ comments below. Changes to the manuscript are highlighted in blue. 

We believe that the revisions prompted by the reviewers’ comments have strengthened our manuscript. We hope it is now suitable for publication and look forward to hearing from you in course. 

Yours faithfully

Reviewers’ Comments to Author:

Reviewer: 1

1.While the article holds certain research value, it falls short in delivering compelling theoretical discussions. A well-developed theoretical foundation is crucial for any academic work. Moreover, the paper tends to make numerous assumptions and addresses a wide array of research questions, lacking the necessary focus. It is suggested that the author refines the research questions for a more coherent presentation. In all, the current manuscript is not well-grounded in the related literature, nor does it develop a theoretical framework that is novel enough for the journal readers.

AUTHORS’ RESPONSE: Thank you very much for your review of our manuscript and very helpful comments. We have made revisions according to your comments. We have added more relevant research and viewpoints to make the theoretical foundation more profound. (pp. 4-7, marked in blue):

Family support as the first mediating variable

Family support refers to the positive resources that individuals acquire from their family relationships [27], a dimension of social support that is a contextual factor known to influence career-related behavior. Many psychologists have emphasized the significance of family support in vocational development because it is directly linked to career search self-efficacy [28]. On the one hand, family support can bring emotional encouragement and work support to individuals. On the other hand, it can relieve personal stress so that they can have more energy to devote to their work [61]. Some studies have suggested that individuals supported by their families report greater work volition. For instance, Founad et al. (2015) developed the Family Influence Scale (FIS) to examine the influence of family on career and work choices. They also found that family's informational support and financial support can impact career and work choices, which can be seen as work volition [15]; in other words, family support can predict a higher level of work volition. Other studies have suggested that family support is significantly related to career optimism[11], which involves making free work choices with a positive mindset despite obstacles, fitting the definition of work volition. According to some studies conducted on Chinese college students, those with adequate career-related parental support are more likely to feel a sense of control, curiosity, confidence about work, and are able to choose the work that best suits their needs, interests, and values. In other words, individuals with greater family support will have greater work volition [29][30][31]. 

Furthermore, research indicates that there is a negative correlation between financial constraints and familial support. This implies that families who are struggling financially are less likely to provide their offspring with monetary assistance or various forms of financial aid. For instance, economic stress can negatively impact parenting styles, where high-quality parenting offers children comprehensive support. Studies have also shown that young individuals from families who are economically disadvantaged tend to encounter both internal and external issues[10], which can be attributed to the lack of familial support due to adverse economic circumstances.

Based on these studies, we establish the first hypothesis：

H1: Family support mediates the relationship between economic constraints and work volition.

Work resources as a second mediating variable

Work resources refer to the factors that stimulate personal growth, help achieve work goals, and mitigate the costs of job demands, which reflect physical, social, psychological, and organizational aspects[34]. They are intricately linked to experiences of job satisfaction, autonomy, purpose, engagement, meaningful work, and job performance[35]. Numerous empirical studies have found that when individuals lose resources at work, they are more likely to have negative physiological outcomes such as burnout and depression[62], which could adversely affects the work volition. According to Barbier et al. (2013), work and personal resources exert a causal effect on job engagement, which is a component of work volition[36]. Furthermore, individuals who lack essential work resources may feel burned out[37], making it more challenging to make informed job decisions, i.e., exhibiting a lower level of work volition. Cheung et al. (2020) examined the relationship between personal resources, constraints, and work volition among undergraduate students in the United States and Hong Kong. Their findings revealed a positive correlation between work resources and work volition, indicating that individuals with greater work resources are more likely to possess the capacity to make informed work decisions[24]. Choi et al. (2022) examined the relationship between social class and work volition, and their findings revealed a positive correlation between the two[38]. This supports the theory that individuals from lower social classes, who may face economic constraints, tend to exhibit lower levels of work volition. A study among Chinese college students revealed that those from high socioeconomic backgrounds, who have greater access to helpful work resources, tend to exhibit higher levels of work volition [39]. Additionally, research has shown that economic constraints can limit individuals' access to work resources. Bourdieu (2008) suggested that economic constraints limit individuals' access to these resources, as individuals or their families may need to pay for them, which can be difficult when facing economic constraints [40]. Duffy et al. (2016) further suggested that economic constraints limit capital resources that can be applied in the workplace, making it more difficult for individuals in economic constraints to acquire the resources that privileged individuals take for granted [17]. A study among labour market entrants found that individuals whose parents had higher education levels, were employed, or were from the service class (indicating greater economic income) had greater access to job-finding resources [41]. In other words, less economic constraint promotes greater access to job resources.

Based on these studies, we establish the second hypothesis：

H2: Work resources mediate the relationship between economic constraints and work volition.

Family support and Work resources

About the relationship between these two mediated variables, some studies indicated the relationship between these two mediating variables has been explored in several studies, which suggest that family support is associated with work resources. Several theories have demonstrated the relationship between family and work resources. Bourdieu's theory (2008) postulates that families with diverse capital can provide individuals with resources, including work-related resources such as a professional network, job opportunities, and a clear understanding of job rules [40]. Conservation of Resources theory raised by Hobfoll indicates that in order to prevent the loss of resources, individuals invest in resources and acquire the corresponding resources[63]. For college students, families make many investments in resources for their work. The family investment model suggests that families of higher socioeconomic status tend to leverage their financial and networking strengths for investment, which means more family support promotes more work resources[64]. Researchers have also conducted empirical studies to test this relationship. Fass et al. (2013) found that families with high occupational status can provide their children with work resources such as professional training, communication skills, and managerial abilities [43]. In other words, individuals with various forms of family support are likely to have access to ample work resources. Wayne and Matthews et al. (2020) conducted a cross-sectional survey of employees and found that work resources are associated with lower work-to-family conflict[44], which suggests that individuals can receive greater family support for their work. Furthermore, a study focusing on college students revealed that higher levels of family cohesion were associated with career thoughts, an important aspect of work volition. This suggests that a harmonious family can provide greater support and promote a higher level of work volition[45].

Based on these studies and combine the findings that family support and work resource play important roles between economic constraints and work volition, we established the third hypothesis:

H3: Family support and work resources played a chain-mediating role in the relationship between economic constraints and work volition.

Reviewer: 2

1.I found some errors in your citations.

AUTHORS’ RESPONSE: Thank you very much for your review of our manuscript and very helpful comments. We have revised the citation format of 30 references and added 4 references. (pp. 17-23, marked in blue):

References

1.Du Yang. The Trend of Changes in Employment of College Students and Suggestions for Countermeasures. People's Tribune. 2022;(17):96-98.

2.Duffy, R. D., Kim, H. J., Boren, S., Pendleton, L., & Perez, G. Lifetime experiences of economic constraints and marginalization among incoming college students: A latent profile analysis. Journal of Diversity in Higher Education. 2023;16(3), 384–396. https://doi.org/10.1037/dhe0000344

3.Wei, J., Chan, S. H. J., & Autin, K. Assessing perceived future decent work securement among Chinese impoverished college students. Journal of Career Assessment. 2022;30(1), 3–22. https://doi.org/10.1177/10690727211005653.

4.Duffy, R. D., Diemer, M. A., Perry, J. C., Laurenzi, C., & Torrey, C. L. The construction and initial validation of the work volition scale. Journal of Vocational Behavior. 2011; 80(2): 400–411. doi:10.1016/j.jvb.2011.04. 002

5.ZHANG Juncheng; HUANG Yingjie; ZHANG Shuying; LIU Dege; QU Shuguang.The Measurement and Individual Differences of College Students’ Work Volition. Psychology:Techniques and Applications. 2019;7(10):629-640.doi:10.16842/j.cnki.issn2095-5588.2019.10.008.

6.Allan, BA., Sterling, HM., Duffy, R.D. Longitudinal relations among economic deprivation, work volition, and academic satisfaction: a psychology of working perspective. International Journal for Educational and Vocational Guidance. 2020; 20:311–329. https://doi.org/10.1007/s10775-019-09405-3.

7.Kim, T., & Allan, B. A. Examining Classism and Critical Consciousness Within Psychology of Working Theory. Journal of Career Assessment. 2021; 29(4):644–660. https://doi.org/10.1177/1069072721998418

8.Wang, Z. Y. and C. K. Li, et al. Family Economic Strain and Adolescent Aggression during the COVID-19 Pandemic: Roles of Interparental Conflict and Parent-Child Conflict. APPLIED RESEARCH IN QUALITY OF LIFE. 2022;17(4): 2369-2385. doi:10.1007/s11482-022-10042-2

9.Sprung, J. M. Economic Stress, Family Distress, and Work-Family Conflict among Farm Couples. JOURNAL OF AGROMEDICINE. 2022;27(2): 154-168. doi:10.1080/1059924X.2021.1944417

10.Conger, R. D., Wallace, L. E., Sun, Y., Simons, R. L., Mcloyd, V. C., & Brody, G. H. Economic pressure in african american families: a replication and extension of the family stress model. Dev Psychol. 2002; 38(2): 179–193.

11.Newman, A. and K. Dunwoodie, et al. Openness to Experience and the Career Adaptability of Refugees: How Do Career Optimism and Family Social Support Matter? JOURNAL OF CAREER ASSESSMENT. 2022;30(2): 309-328. doi: 10.1177/10690727211041532

12.Agrawal, S. and S. Singh. Predictors of subjective career success amongst women employees: moderating role of perceived organizational support and marital status. GENDER IN MANAGEMENT. 2022;37(3): 344-359. DOI10.1108/GM-06-2020-0187

13.Mehreen, A. and Z. Ali. Really shocks can't be ignored: the effects of career shocks on career development and how family support moderates this relationship? 2022;INTERNATIONAL JOURNAL FOR EDUCATIONAL AND VOCATIONAL GUIDANCE. doi:10.1007/s10775-022-09574-8 

14.Yang, D. L. and G. X. Fang, et al. Impact of work-family support on job burnout among primary health workers and the mediating role of career identity: A cross-sectional study. FRONTIERS IN PUBLIC HEALTH 11. 2023. doi:10.3389/fpubh.2023.1115792

15. Fouad, N. A. , Cotter, E. W. , Fitzpatrick, M. E. , Kantamneni, N. , Carter, L. , & Bernfeld, S. Development and validation of the family influence scale. Journal of Career Assessment. 2010；18(3), 276-291.

16.Pursio, H. and A. Siukola, et al. Associations between Work Resources and Work Ability among Forestry Professionals. SUSTAINABILITY. 2021;13(9). doi:10.3390/su13094822

17.Duffy, R. D., Blustein, D. L., Diemer, M. A., and Autin, K. L. The psychology of working theory. J. Counsel. Psychol. 2016; 63:127–148. https://doi.org/10.1037/cou0000140

18.Hai, L., Bao, X., & Li, W. Factors associated with work volition among Chinese undergraduates. Front. Psychol. 2022; 13:1037185. doi:10.3389/fpsyg. 2022.1037185

19.Zhao, F., Wang, Y. & Ping, L. Work volition and academic satisfaction in Chinese female college students: the moderating role of incremental theory of work volition. Int J Educ Vocat Guidance. 2022. https://doi.org/10.1007/s10775-022-09542-2

20.Kim, N.R., Kim, H.J., & Lee, K.H. Social Support and Occupational Engagement Among Korean Undergraduates: The Moderating and Mediating Effect of Work Volition. Journal of Career Development. 2018; 45:285–298.

21.Kim, N.R., Kim, H. J., & Lee, K.H. Social status and decent work: test of a moderated mediation model. The Career Development Quarterly. 2020; 68(3):208–221. https://doi.org/10.1002/cdq.12232

22.Brief, A. P., Konovsky, M. A., Goodwin, R., & Link, K. Inferring the meaning of work from the effects of unemployment. Journal of Applied Social Psychology. 1995; 25(8): 693–711. https://doi. org/10.1111/j.1559-1816. 1995.tb01769.x

23.Duffy, R. D., Douglass, R. P., Autin, K. L., & Allan, B. A. . Examining predictors of work volition among undergraduate students. Journal of Career Assessment. 2016; 24(3), 441–459. doi:10.1177/1069072715599377

24.Cheung, F., Ngo, H. Y., & Leung, A. Predicting work volition among undergraduate students in the united states and Hong Kong. Journal of Career Development. 2020; 47(5): 565–578. https://doi.org/10.1177/0894845318803469

25.Tokar, D. M., & Kaut, K. P. Predictors of decent work among workers with Chiari malformation: an empirical test of the psychology of working theory. Journal of Vocational Behavior. 2018; 106(JUN.), 126–137.

26.Autin, K. L., Douglass, R. P., Duffy, R. D., England, J. W., & Allan, B. A. Subjective social status, work volition, and career adaptability: a longitudinal study. Journal of Vocational Behavior. 2017; 99:1–10. https://doi.org/10.1016/j.jvb.2016.11.007

27.Betz, N. E. Implications of the null environment hypothesis for women’s career development and for counseling psychology. The Counseling Psychologist. 1989; 17, 136–144. doi:10.1177/0011000089171008

28.Nota, L., Ferrari, L., Solberg, V. S. H., & Soresi, S. Career search self-efficacy, family support, and career indecision with Italian youth. Journal of Career Assessment. 2007; 15(2), 181–193.

29.Zeng, Q. and J. Li, et al. How does career-related parental support enhance career adaptability: the multiple mediating roles of resilience and hope. CURRENT PSYCHOLOGY. 2023;42(29): 25193-25205. doi:10.1007/s12144-022-03478-0

30.Zhang, J. H. and M. Y

---

## [Decision Letter · Decision Letter 2]

28 May 2024

PONE-D-23-23939R2Effect of Family Support and Work Resources in the Relationship of Economic Constraints and Work Volition : Evidence from ChinaPLOS ONE

Dear Dr. Wang,

Thank you for submitting your manuscript to PLOS ONE. After careful consideration, we feel that it has merit but does not fully meet PLOS ONE’s publication criteria as it currently stands. Therefore, we invite you to submit a revised version of the manuscript that addresses the points raised during the review process.

We look forward to receiving your revised manuscript.

Kind regards,

Henri Tilga, PhD

Academic Editor

PLOS ONE

Additional Editor Comments:

The Reviewers point out several concerns related to this manuscript. Please revise the manuscript as suggested by the Reviewers.

Reviewers' comments:

Reviewer's Responses to Questions

**Comments to the Author**

1. If the authors have adequately addressed your comments raised in a previous round of review and you feel that this manuscript is now acceptable for publication, you may indicate that here to bypass the “Comments to the Author” section, enter your conflict of interest statement in the “Confidential to Editor” section, and submit your "Accept" recommendation.

Reviewer #1: (No Response)

Reviewer #2: All comments have been addressed

2. Is the manuscript technically sound, and do the data support the conclusions?

Reviewer #1: Partly

Reviewer #2: Yes

3. Has the statistical analysis been performed appropriately and rigorously? 

Reviewer #1: I Don't Know

Reviewer #2: Yes

4. Have the authors made all data underlying the findings in their manuscript fully available?

Reviewer #1: Yes

Reviewer #2: No

5. Is the manuscript presented in an intelligible fashion and written in standard English?

Reviewer #1: Yes

Reviewer #2: Yes

6. Review Comments to the Author

Reviewer #1: This paper is related to an interesting topic, namely Effect of Family Support and Work Resources in the Relationship of Economic Constraint and Work Volition : Evidence from China for PLOS ONE

However, the manuscript needs to be further developed in order to meet the expected academic requirements.

The introductory section of the document lacks the requisite depth and detail. It is essential to provide a more comprehensive introduction, one that not only sets the stage for the study but also explicitly identifies the existing knowledge gaps that this research aims to address. Without a clear articulation of these gaps in the introduction, the readers may find it challenging to understand the context and significance of the research.

The literature review is weak. The authors must explain the main theories in the field of the study and identify the gaps in previous models. By doing so, researchers can develop a more compelling and theoretically sound model that addresses the shortcomings of earlier studies.

The segment dedicated to the literature review and hypotheses development exhibits some weaknesses. To bolster the quality of this section, it is imperative to delve into previous theories and research within the field. This entails a thorough discussion of the existing body of knowledge, highlighting key theories, findings, and their implications. Furthermore, it is crucial to elucidate the novel contribution of this research and provide a comprehensive explanation of how the model underpinning the study was constructed. The absence of this information leaves the model without a solid foundation, undermining the credibility and robustness of the research.

The summary or conclusion of the study appears to be overly succinct and leaves some key elements unaddressed. A comprehensive conclusion should not only summarize the key findings but also provide answers to the research questions posed at the outset of the study. By failing to address these questions in the conclusion, the research may be perceived as incomplete and lacking in closure. A more comprehensive summary and conclusion would help readers to better grasp the implications and significance of the study's findings.

Reviewer #2: Dear Authors

Thank you for addressing comments given from my end.

One little correction is still needed. I found that the citation issue still prevails. Please correct it.

Sincerely.

7. PLOS authors have the option to publish the peer review history of their article (what does this mean?). If published, this will include your full peer review and any attached files.

Reviewer #1: No

Reviewer #2: No

---

## [Author Response · Author response to Decision Letter 2]

10 Aug 2024

PONE-D-23-23939

Effect of Family Support and Work Resources in the Relationship of Economic Constraints and Work Volition : Evidence from China

PLOS ONE

Dear Dr. Henri Tilga

Thank you very much for the opportunity to revise our manuscript entitled “Effect of Family Support and Work Resources in the Relationship of Economic Constraint and Work Volition : Evidence from China”. We would like to thank you and the reviewers for their helpful comments. The manuscript has been carefully revised according to the comments raised by the reviewers. Please find a point-to-point response to reviewers’ comments below. Changes to the manuscript are highlighted in blue. 

We believe that the revisions prompted by the reviewers’ comments have strengthened our manuscript. We hope it is now suitable for publication and look forward to hearing from you in course. 

Yours faithfully

Reviewers’ Comments to Author:

Reviewer: 1

1.The introductory section of the document lacks the requisite depth and detail. It is essential to provide a more comprehensive introduction, one that not only sets the stage for the study but also explicitly identifies the existing knowledge gaps that this research aims to address. Without a clear articulation of these gaps in the introduction, the readers may find it challenging to understand the context and significance of the research. The literature review is weak. The authors must explain the main theories in the field of the study and identify the gaps in previous models. By doing so, researchers can develop a more compelling and theoretically sound model that addresses the shortcomings of earlier studies.

AUTHORS’ RESPONSE: Thank you very much for your review of our manuscript and very helpful comments. We have fleshed out the introduction, particularly enriched the literature review. Moreover, we have detailed the contributions and shortcomings of previous research. Previous research has not fully considered the mediating role of objective factors in the relationships between economic constraints and work volition, such as family support and work resources. Therefore, it has not explored the internal mechanisms that economic constraints use to influence work volition through family support and work resources have not been fully explored.(pp. 3-4, marked in blue):

Work volition is defined as an individual's sense of agency or freedom to make a career choice [4], highlighting the potential of subjective belief to overcome external constraints. For contemporary Chinese college students, their work volition exhibits two key characteristics. Firstly, overall, the level of work volition is generally low. Secondly, individuals from low-income families possess lower work volition than those from high socioeconomic status families [5]. Notably, economic constraints can negatively impact work volition, a finding consistently supported by some researches in different areas. It was found that individuals with greater economic constraints tend to have lower levels of career decision-making among university students and employees [6][7]. Meanwhile, The PWT (Duffy et al., 2016) was developed to capture the work lives of all individuals. In PWT moedel, economic constraints and economic constraints are positioned as key predictors, in particular, economic constraints can objectively predict work volition. Additionally, economic constraints can also escalate conflicts and distress within families [8][9], for example, young individuals from families who are economically disadvantaged tend to encounter both internal and external issues [10], undermining adolescents' access to family support. Given the strong emphasis on family values in Chinese culture, family support also plays a crucial role in individuals' growth and career development. Family support can bring emotional encouragement and work support to individuals, so that they feel more confident when they are making choice about work[11][12][13][14][15]. Furthermore, studies suggest that economic constraints are also associated with work resources [15], for instance, Duffy et al. (2016) suggested that economic constraints limit capital resources that can be applied in the workplace, making it more difficult for individuals in economic constraints to acquire the resources that privileged individuals take for granted [16]. And work resources are recognised as a fundamental requirement for individuals in career development [17].

Previous studies have examined the relationships among economic constraints, family support, work resources and work volition. It was demonstrated the direct effects of economic constraints on work volition and the direct impact of family support and work resources on work volition. However, there are still knowledge gaps. Obviously previous research has not fully considered the mediating role of objective factors in the relationships between economic constraints and work volition, such as family support and work resources. Therefore, it has not explored the internal mechanisms that economic constraints use to influence work volition through family support and work resources have not been fully explored.

2.The segment dedicated to the literature review and hypotheses development exhibits some weaknesses. To bolster the quality of this section, it is imperative to delve into previous theories and research within the field. This entails a thorough discussion of the existing body of knowledge, highlighting key theories, findings, and their implications. Furthermore, it is crucial to elucidate the novel contribution of this research and provide a comprehensive explanation of how the model underpinning the study was constructed. The absence of this information leaves the model without a solid foundation, undermining the credibility and robustness of the research.

AUTHORS’ RESPONSE: Thank you very much for your review of our manuscript and very helpful comments. We have strengthened literature review and introduced the relevant theory, such as the PWT put forward by Duffy, which proved that economic constraints can objectively predict work volition. And we have provided a comprehensive explanation of how the model underpinning the study was constructed.(pp.8 marked in blue):

The aim of this study was to examine the mechanisms underlying the relationship between economic constrains and work volition through the family support and work resources using a mediation model. In this research, three variables are positioned as key predictors of work volition: (a) economic constraints and (b) family support and (c) work resource as variables that both predict work volition and explain the mediating role of family support and work resource. Specifically, (1) family support mediates the relationship between economic constraints and work volition; (2) Work resources mediate the relationship between economic constraints and work volition; (3) Family support and work resources played a chain-mediating role in the relationship between economic constraints and work volition. Based on these three hypothesis pathways, we constructed the model. (See Figure1). 

3.The summary or conclusion of the study appears to be overly succinct and leaves some key elements unaddressed. A comprehensive conclusion should not only summarize the key findings but also provide answers to the research questions posed at the outset of the study. By failing to address these questions in the conclusion, the research may be perceived as incomplete and lacking in closure. A more comprehensive summary and conclusion would help readers to better grasp the implications and significance of the study's findings.

AUTHORS’RESPONSE: Thank you very much for your review of our manuscript and very helpful comments. We have added the summary and conclusion to make it more helpful for readers to better grasp the research. (pp.2,15 marked in blue):

Abstract

Background 

As the pace of economic development slows, college students are facing an increasingly challenging employment landscape. For instance, the expansion of higher education has led to a swell in the number of job seekers, which has in turn intensified competition. Given the limited job opportunities, it's understandable that many college students are developing a pessimistic employment mindset. Therefore, it's crucial to explore how objective factors influence their work aspirations. But few studies have explored the role of mediating factors between the two, such as family and resource factors.Thus, this study examines the effects of family support and work resources between the relationship between economic constraints and work volition. 

Methods

The study examined 1249 Chinese undergraduate students as participants ((714 men and 535 women; Mage = 19.32, SD = 1.50), using the questionnaire with the Wenjuanxing online survey tool. The questionnaire were collected between August, 2022 and December, 2022. SPSS 21.0 and AMOS 24.0 were used to conducted a comprehensive analysis of the collected data, and investigate the relationships among latent variables and assess the goodness of fit of the observed indicators on their associated latent constructs. Additionally, we evaluated all the hypothesized direct and indirect effects.

Results

The results showed the direct and indirect relationships among economic constraints, family support, work resources and work volition. Economic constraints can directly predict work volition. Moreover, economic constraints have a significant negative impact on work volition via two mediators: family support and work resources. On the one hand, economic constraints negatively affect work volition through family support and work resource separately. On the other hand, economic constraints negatively predict family support and work resource, thus negatively impact work volition. 

Contribution

The current study has established the independent mediating and chain-mediated effects of family support and work resources on the relationship between economic constraints and work volition. This deeper understanding of internal mechanisms provides valuable insights that can inform strategies for enhancing individual's work volition, particularly from the perspectives of economic constraints, family support, and work volition.

Conclusion

The present study, which focused on Chinese college students, delved into the relationship between economic constraints and work volition, taking into account the mediating effects of family support and work resources. The findings confirmed the mediating roles and chain mediating roles of these factors in the relationship. Specifically, three indirect pathways were identified: family support mediates the relationship between economic constraints and work volition; work resources mediate the relationship between economic constraints and work volition; family support and work resources played a chain-mediating role in the relationship between economic constraints and work volition. While acknowledging several limitations, the study provides valuable directions for future research on the importance of family support and work resources in career development. It also underscores the need for schools, families, and society to recognize and act on these factors to promote positive career outcomes for students.

Reviewer: 2

1.One little correction is still needed. I found that the citation issue still prevails. Please correct it.

AUTHORS’ RESPONSE: Thank you very much for your review of our manuscript and very helpful comments. We have corrected the citation issue in the text and endnotes. (pp. 3-23, marked in blue):

References

1.Du Y. The Trend of Changes in Employment of College Students and Suggestions for Countermeasures. People's Tribune. 2022;17:96-98.

2.Duffy, R. D., Kim, H. J., Boren, S., Pendleton, L., & Perez, G. Lifetime experiences of economic constraints and marginalization among incoming college students: A latent profile analysis. Journal of Diversity in Higher Education. 2023;16(3),:384–396. https://doi.org/10.1037/dhe0000344

3.Wei, J., Chan, S. H. J., & Autin, K. Assessing perceived future decent work securement among Chinese impoverished college students. Journal of Career Assessment. 2022;30(1):3–22. https://doi.org/10.1177/10690727211005653.

4.Duffy, R. D., Diemer, M. A., Perry, J. C., Laurenzi, C., & Torrey, C. L. The construction and initial validation of the work volition scale. Journal of Vocational Behavior. 2011; 80(2): 400–411. doi:10.1016/j.jvb.2011.04. 002

5.Zhang J, Huang Y J, Zhang S Y, Liu D G;, Qu S G. The Measurement and Individual Differences of College Students’ Work Volition. Psychology:Techniques and Applications. 2019;7(10):629-640.doi:10.16842/j.cnki.issn2095-5588.2019.10.008.

6.Allan, BA., Sterling, HM., Duffy, R.D. Longitudinal relations among economic deprivation, work volition, and academic satisfaction: a psychology of working perspective. International Journal for Educational and Vocational Guidance. 2020; 20:311–329. https://doi.org/10.1007/s10775-019-09405-3.

7.Kim, T., & Allan, B. A. Examining Classism and Critical Consciousness Within Psychology of Working Theory. Journal of Career Assessment. 2021; 29(4):644–660. https://doi.org/10.1177/1069072721998418

8.Wang, Z. Y. and C. K. Li, et al. Family Economic Strain and Adolescent Aggression during the COVID-19 Pandemic: Roles of Interparental Conflict and Parent-Child Conflict. APPLIED RESEARCH IN QUALITY OF LIFE. 2022;17(4): 2369-2385. doi:10.1007/s11482-022-10042-2

9.Sprung, J. M. Economic Stress, Family Distress, and Work-Family Conflict among Farm Couples. JOURNAL OF AGROMEDICINE. 2022;27(2): 154-168. doi:10.1080/1059924X.2021.1944417

10.Newman, A. and K. Dunwoodie, et al. Openness to Experience and the Career Adaptability of Refugees: How Do Career Optimism and Family Social Support Matter? JOURNAL OF CAREER ASSESSMENT. 2022;30(2): 309-328. doi: 10.1177/10690727211041532

11.Agrawal, S. and S. Singh. Predictors of subjective career success amongst women employees: moderating role of perceived organizational support and marital status. GENDER IN MANAGEMENT. 2022;37(3): 344-359. DOI10.1108/GM-06-2020-0187

12.Mehreen, A. and Z. Ali. Really shocks can't be ignored: the effects of career shocks on career development and how family support moderates this relationship? 2022;INTERNATIONAL JOURNAL FOR EDUCATIONAL AND VOCATIONAL GUIDANCE. doi:10.1007/s10775-022-09574-8 

13.Yang, D. L. and G. X. Fang, et al. Impact of work-family support on job burnout among primary health workers and the mediating role of career identity: A cross-sectional study. FRONTIERS IN PUBLIC HEALTH 11. 2023. doi:10.3389/fpubh.2023.1115792

14. Conger, R. D., Wallace, L. E., Sun, Y., Simons, R. L., Mcloyd, V. C., & Brody, G. H. Economic pressure in african american families: a replication and extension of the family stress model. Dev Psychol. 2002; 38(2): 179–193.

15.Fouad, N. A. , Cotter, E. W. , Fitzpatrick, M. E. , Kantamneni, N. , Carter, L. , & Bernfeld, S. Development and validation of the family influence scale. Journal of Career Assessment. 2010；18(3), 276-291.

16.Duffy, R. D., Blustein, D. L., Diemer, M. A., and Autin, K. L. The psychology of working theory. J. Counsel. Psychol. 2016; 63:127–148. https://doi.org/10.1037/cou0000140

17.Pursio, H. and A. Siukola, et al. Associations between Work Resources and Work Ability among Forestry Professionals. SUSTAINABILITY. 2021;13(9). doi:10.3390/su13094822

18.Hai, L., Bao, X., & Li, W. Factors associated with work volition among Chinese undergraduates. Front. Psychol. 2022; 13:1037185. doi:10.3389/fpsyg. 2022.1037185

19.Zhao, F., Wang, Y. & Ping, L. Work volition and academic satisfaction in Chinese female college students: the moderating role of incremental theory of work volition. Int J Educ Vocat Guidance. 2022. https://doi.org/10.1007/s10775-022-09542-2

20.Kim, N.R., Kim, H.J., & Lee, K.H. Social Support and Occupational Engagement Among Korean Undergraduates: The Moderating and Mediating Effect of Work Volition. Journal of Career Development. 2018; 45:285–298.

21.Kim, N.R., Kim, H. J., & Lee, K.H. Social status and decent work: test of a moderated mediation model. The Career Development Quarterly. 2020; 68(3):208–221. https://doi.org/10.1002/cdq.12232

22.Brief, A. P., Konovsky, M. A

---

## [Editor Report · Decision Letter 3]

30 Aug 2024

Effect of Family Support and Work Resources in the Relationship of Economic Constraints and Work Volition : Evidence from China

PONE-D-23-23939R3

Dear Dr. Wang,

We’re pleased to inform you that your manuscript has been judged scientifically suitable for publication and will be formally accepted for publication once it meets all outstanding technical requirements.

Kind regards,

Henri Tilga, PhD

Academic Editor

PLOS ONE
---

## [Editor Report · Acceptance letter]

9 Sep 2024

PONE-D-23-23939R3 

PLOS ONE

Dear Dr. Wang, 

I'm pleased to inform you that your manuscript has been deemed suitable for publication in PLOS ONE. Congratulations! Your manuscript is now being handed over to our production team.

Kind regards, 

on behalf of

Dr. Henri Tilga 

Academic Editor

PLOS ONE